# Urolithins: Diet-Derived Bioavailable Metabolites to Tackle Diabetes

**DOI:** 10.3390/nu13124285

**Published:** 2021-11-27

**Authors:** Ana F. Raimundo, Sofia Ferreira, Francisco A. Tomás-Barberán, Claudia N. Santos, Regina Menezes

**Affiliations:** 1iBET, Instituto de Biologia Experimental e Tecnológica, Apartado 12, 2781-901 Oeiras, Portugal; ana.raimundo@nms.unl.pt; 2ITQB-NOVA, Instituto de Tecnologia Química e Biológica António Xavier, Universidade Nova de Lisboa, Av. da República, 2781-157 Oeiras, Portugal; 3CEDOC, Chronic Diseases Research Center, NOVA Medical School, Faculdade de Ciências Médicas, Universidade NOVA de Lisboa, Campo dos Mártires da Pátria, 130, 1169-056 Lisboa, Portugal; sofia.ferreira@nms.unl.pt (S.F.); claudia.nunes.santos@nms.unl.pt (C.N.S.); 4CBIOS—Universidade Lusófona’s Research Center for Biosciences & Health Technologies, Campo Grande 376, 1749-024 Lisboa, Portugal; 5Department of Food Science and Technology, CEBAS-CSIC, Campus Universitario de Espinardo, edf 25, 30100 Murcia, Spain; fatomas@cebas.csic.es

**Keywords:** diabetes, metabotypes, (poly)phenols, urolithins

## Abstract

Diabetes remains one of the leading causes of deaths and co-morbidities in the world, with tremendous human, social and economic costs. Therefore, despite therapeutics and technological advancements, improved strategies to tackle diabetes management are still needed. One of the suggested strategies is the consumption of (poly)phenols. Positive outcomes of dietary (poly)phenols have been pointed out towards different features in diabetes. This is the case of ellagitannins, which are present in numerous foodstuffs such as pomegranate, berries, and nuts. Ellagitannins have been reported to have a multitude of effects on metabolic diseases. However, these compounds have high molecular weight and do not reach circulation at effective concentrations, being metabolized in smaller compounds. After being metabolized into ellagic acid in the small intestine, the colonic microbiota hydrolyzes and metabolizes ellagic acid into dibenzopyran-6-one derivatives, known as urolithins. These low molecular weight compounds reach circulation in considerable concentrations ranging until micromolar levels, capable of reaching target tissues. Different urolithins are formed throughout the metabolization process, but urolithin A, isourolithin A, and urolithin B, and their phase-II metabolites are the most frequent ones. In recent years, urolithins have been the focus of attention in regard to their effects on a multiplicity of chronic diseases, including cancer and diabetes. In this review, we will discuss the latest advances about the protective effects of urolithins on diabetes.

## 1. Diabetes Mellitus

Since 1965, the World Health Organization (WHO) has occasionally updated guidance on how to classify diabetes mellitus (DM). According to the current definition, DM comprises a group of disorders identified by the presence of hyperglycemia in the absence of treatment [1]. Exhibiting a heterogeneous etiopathology, DM is a multifaceted disorder simplistically characterized as faults in insulin secretion and/or sensitivity, and disturbances of carbohydrate, lipid, and protein metabolism [2,3,4].

DM is an epidemic with frightening numbers worldwide, constituting one of the top 10 causes of death in adults [5]. In 2019, it was estimated the existence of 351.7 million people between 20 and 64 years of age with diagnosed or undiagnosed DM. Without interventions to effectively halt the increase in the disease, this number is expected to account for 486.1 million by 2045 [6]. Noteworthy, even though therapies have come a long way in the last decades, with ground-breaking treatments that allow individuals to live longer and better, there is still a 10-year reduction in life expectancy for diabetic individuals. In fact, the numbers do not deceive and reveal an urgent need to provide better solutions for DM management.

Subtyping DM is important in clinical care for diagnosis, to guide treatment decisions. Conventionally, most DM cases fall into two broad pathogenic categories: Type 1 diabetes mellitus (T1DM) and Type 2 diabetes mellitus (T2DM). However, in some people, this strict classification is not applicable due to the involvement of other genetic, immunological, and neuroendocrinological factors in the disease pathogenesis. Based on that, new classifications beyond the two typical types start to be considered, including hybrid forms of diabetes, monogenic diabetes, unclassified diabetes, pregnancy-related diabetes, among others [1]. 

### 1.1. β-Cell Dysfunction and Insulin Resistance

T2DM, hereafter referred to as diabetes, is the most common form of the disease, accounting for around 90% of all diabetes cases [1]. It is most commonly diagnosed in older adults, however, its prevalence is increasing in children, adolescents, and younger adults due to the rising levels of obesity, sedentarism, and poor diet habits [7]. 

Insulin resistance is the earliest detectable alteration in individuals who are likely to develop diabetes, preceding the clinical diagnosis of the disease by up to 15 years [2,8]. It can occur in multiple organs and tissues, such as muscle, liver, adipose tissue, kidney, gastrointestinal tract, vasculature/brain tissues, and pancreatic β-cells [9,10,11,12]. Insulin resistance has been associated with increased serine phosphorylation of insulin receptor substrates (IRS), which are involved in the phosphatidylinositol 3-kinase (PI3K) pathway [2,13]. At the time of ligand binding, insulin receptors change their conformation and autophosphorylate, leading to the recruitment and phosphorylation of Shc proteins and IRS. Shc activates the Ras-MAPK pathway, whereas phosphorylated IRS proteins activate the intracellular signaling PI3K molecule. Once activated, PI3K induces glucose transporter type 4 (GLUT4) translocation to the plasma membrane, thus mediating glucose uptake into skeletal muscle [14]. Increased levels of phosphorylated IRS inhibit the initial phosphorylation of insulin receptors and increases IRS degradation, further contributing to insulin resistance state [2]. 

At an initial phase, when cells are resistant to insulin, pancreatic β-cells produce high amounts of this hormone, leading to the increase in its levels in the blood (hyperinsulinemia). Along with its overexpression, reduced insulin clearance also appears to contribute to this state. Concerning this, lower activity of insulin-degrading enzymes in the liver was recently associated with higher plasma insulin levels in a group of individuals at high risk of developing T2DM [15]. At later stages, with a gradual decline of pancreatic β-cell functionality, these cells become ineffective in producing and secreting insulin in sufficient levels to keep up with increased insulin demand. At that moment, hyperglycemia is installed [9].

As a consequence of persistent hyperglycemia, co-morbidities may occur ultimately leading to end-organ failure, especially retina, kidney, and nerves. Adding to this, people with diabetes also have an increased risk to develop other pathologies including heart, peripheral arterial, and cerebrovascular disease, obesity, erectile dysfunction, and non-alcoholic fatty liver disease [1]. Another consequence of chronic high blood glucose is the production and accumulation of advanced glycation end-products (AGE). These result from the non-enzymatic reaction of sugars with proteins or lipids. The accumulation of AGE in the blood and organs results in microvascular complications, which are also implicated in Alzheimer’s disease, Parkinson’s disease, and atherosclerosis [16].

Besides carbohydrates metabolism alterations, diabetes is also linked to severe abnormalities in lipids and lipoproteins. Although the exact mechanisms are not yet fully elucidated, mounting evidence has shown that insulin resistance contributes to the development of diabetic dyslipidemia [17]. In that regard, increased intracellular lipid levels have been linked to reduced insulin sensitivity in the liver and skeletal muscle [18]. In the liver, excessive accumulation of diacylglycerol (DAG) and ceramide species, both toxic lipid metabolites, modulates the PI3K pathway leading to hepatic insulin resistance, impaired activation of glycogen synthesis, and suppression of gluconeogenesis. Ceramide and other sphingolipids metabolites are also suggested to disrupt pancreatic β-cell function and vascular reactivity [18,19]. Under peripheral insulin resistance state, the free fatty acids flux from lipolysis in adipose tissue increases. Free fatty acids are posteriorly taken up by the liver, leading to the hepatic overproduction of triglycerides, very low-density lipoprotein cholesterol (VLDL), and ApoB [20]. When present in elevated amounts, triglycerides can accumulate and deposit in organs and tissues such as muscle, liver, heart, and pancreas [17]. Hepatic lipase activity also increases, leading to a reduction in HDL cholesterol and to the formation of smaller and denser LDL particles which are associated with a higher rate of nephropathy [17,21]. Features of dyslipidemia can occur many years before the onset of clinically relevant hyperglycemia. They are associated with increased cardiovascular risk [17], which is of utmost importance since diabetic individuals have a two-fold excess risk of vascular diseases [22]. 

Multiple factors have been pointed out as potential contributors for β-cell dysfunction in diabetes, including aging, glucolipotoxicity, islet cholesterol accumulation, islet inflammation, genetic defects in insulin secretion, endocrine disorders, islet amyloid polypeptide (IAPP or amylin) aggregation, among others [23,24,25,26]. 

### 1.2. Chronic Inflammation and Diabetes 

An increasing body of evidence suggests a potential role for inflammation in diabetes pathogenesis. In that regard, several components of the immune system were reported to be altered during the disease, with the most apparent changes occurring in the liver, adipose tissue, pancreatic islets, vasculature, and in circulating leukocytes [26].

When present in excessive amounts, glucose and free fatty acids stress the pancreatic islets and insulin-sensitive tissues, leading to the local production and secretion of pro-inflammatory factors [26]. Circulating levels of acute-phase proteins, such as C-reactive protein (CRP), fibrinogen, haptoglobin, and sialic acid, were reported to be augmented in individuals with diabetes [26,27,28]. In addition, diverse cytokines and chemokines were also shown to be altered, with elevated levels of Interleukin-1β (IL-1β) and IL-6 constituting a predictive marker for the disease progression [28,29,30]. IL-6 was described to be an inducer of apoptosis in pancreatic islets, while inflammasome/IL-1β signaling was considered the most important pathway activated in islets of multiple diabetes models where β-cell damage is observed [30,31]. Interestingly, under high glucose conditions, human pancreatic islets were shown to increase IL-1β and reduce the expression of its respective receptor antagonist, contributing to the establishment of a pro-inflammatory milieu [32]. Through a mechanism of self-stimulation, IL-1β regulates its own production, leading to a vicious cycle that prolongs inflammation, increases nitric oxide (NO) levels, and reduces ATP concentration in the mitochondria–two important parameters involved in β-cell dysfunction and impaired insulin secretion [30]. 

It is also assumed that pro-inflammatory cytokines can induce insulin resistance through the activation of downstream kinases, such as IκB kinase-β (IKKβ), JUN N-terminal kinase (JNK) and p38 MAPK [26]. Corroborating this idea, the inhibition of the IKKβ–nuclear factor-κB (NF-κB) pathway via pharmacological and genetic means ameliorated insulin sensitivity in mice [26] and improved glycemic control in individuals with diabetes [33]. Overproduction of TNF-α in adipose tissue also appeared to exacerbate inflammation and β-cell death in pancreatic islets, and cause insulin resistance in peripheral tissues [30,34]. 

Immune cell infiltration is also a well-documented phenomenon that contributes to islet inflammation. Analysis of histological pancreatic sections has shown high numbers of macrophage marker CD68+ cells in the vicinity of islets derived from individuals with diabetes. Noteworthy, in other studies, the presence of these cells in islets was positively correlated with the occurrence of amyloid structures and fibrosis [26,35] as well as increased body mass index, fasting C-peptide immunoreactivity, fat cell area, and hyperglycemia [36]. In the reported amyloid-positive cases, CD68^+^ cells predominated over tissue repair-oriented macrophages (double-positive for CD163 and CD204 markers) [35]. Such findings support the idea that, during islet inflammation, the macrophage population shifts their polarity toward a more inflammatory phenotype. This switch has been shown to interfere with the initiation and amplification of islet inflammation, contributing to β-cell dysfunction in diabetes mouse models [37]. Increased amounts of pro-inflammatory cells, such as activated macrophages, T helper 1 (TH1), TH17, and CD8^+^ T cells, have also been described to be present in adipose tissue of individuals and animals in insulin-resistant states [38,39]. There, macrophage infiltration stimulates lipolysis and augments IL-6 levels, promoting hepatic gluconeogenesis and insulin resistance [40]. 

In pancreatic islets from high fat-fed human IAPP transgenic mice, amyloid formation was shown to induce the synthesis of interleukins and other pro-inflammatory factors that activate macrophages in vivo [40]. Activation of the receptor for AGEs (RAGE) and FAS receptor were also suggested as potential mediators for IAPP-induced toxicity [41,42], highlighting the importance of amyloid formation in islet inflammation. 

Although it is well established that insulitis is part of the etiopathology of diabetes, there are still important issues that need to be addressed, including: (a) the origin of enhanced islet immune cells (i.e., to understand if they are recruited to the islet or are derived from resident immune cell population); (b) the physiological signaling function of cytokines and chemokines; and (c) the unraveling of cellular sources within the islet responsible for the production of these inflammatory factors.

### 1.3. Islet Amyloid Pathology

Pancreatic amyloid plaques of IAPP have been found in about 90% of individuals with diabetes, mainly as extracellular deposits in the vicinity of β-cells [43]. IAPP is a highly amyloidogenic 37-amino acid peptide concomitantly expressed, processed, and secreted with insulin by pancreatic β-cells in response to glucose intake. Physiologically, IAPP and insulin act in a synergistic way to regulate glucose homeostasis and metabolism [44,45].

Under pre-diabetes and diabetes scenarios, the high demand for insulin production is accompanied by an increase in IAPP expression [46,47]. Consequently, with insulin and IAPP overproduction, and accompanying factors not yet fully understood, the β-cell processing machinery becomes overloaded leading to the accumulation and aggregation of unprocessed and mature IAPP forms [48,49,50]. Although it is agreed that amyloid aggregation processes result in injury of β-cells, the precise cytotoxic mechanisms by which IAPP contributes to diabetes progression and exacerbation remains to be fully elucidated. In early studies, the formation of islet amyloid was reported to be strongly related to a loss of nearly 50% of the β-cell mass and reduced levels of insulin secretion [51]. When located in the interface of β-cells and capillary endothelial cells, IAPP fibrils could impair the normal flow of glucose, thus affecting the release of insulin from the β-cells secretory vesicles [52]. IAPP fibrils were also shown to directly contact the cell surface leading to islet cell damage and apoptosis by inducing the formation of protuberances in the plasma membrane, chromatin condensation, and DNA fragmentation [53]. The current perspectives, however, push forward the soluble IAPP oligomers and protofibrils as the main cytotoxic effectors of β-cell depletion and diabetes onset [54]. Pre-amyloid IAPP oligomers can form ionic pores that leakage the membrane and allow the dysregulated entry of calcium ions into the intracellular environment, thus leading to the activation of apoptosis via caspase and Janus kinase pathways [52,55]. Other mechanisms involved in IAPP-mediated β-cell damage include endoplasmic reticulum (ER) stress, deregulation of ER-associated protein degradation (ERAD) and unfolded protein response (UPR), local inflammation, mitochondrial dysfunction, and oxidative stress [56]. Interestingly, AGE products have been also reported to contribute to IAPP cytotoxicity [57]. 

### 1.4. Role of Autophagy in Β-Cell Function and Survival

Autophagy is referred to as a catabolic process involved in the intracellular clearance of cytotoxic proteins and damaged organelles under stress conditions [58]. Constitutive autophagy has been reported to regulate pancreatic islet architecture, function, and homeostasis. It promotes β-cells survival in conditions conducive to cell death, such as nutrient depletion, hypoxia, and mitochondrial damage [59]. In addition, it also acts as an important protective mechanism against oxidative stress on insulin target tissues [7], through the degradation of the key inhibitor of the antioxidant response, KEAP [60]. 

In this sense, impairment of autophagic pathways in β-cells has been linked to insulin resistance and diabetes onset and progression [58]. Autophagy deficient mutant mice lacking *ATG7* gene have been described to develop a significant degeneration of islets and abnormal insulin secretion. When submitted to a high-fat diet, these animals were unable to stimulate β-cell autophagy unlike their control counterparts and showed an impaired glucose tolerance due to the failure of compensatory hyperplasia of β-cells [61]. Similarly, repression of β-cell autophagy via *Tsc-2* knockout/hyperactivation of the autophagy upstream negative regulator mTORC1 led to increased mitochondrial oxidation and ER stress, which culminated in β-cell failure in mice [62]. mTORC1 is a master kinase responsible for controlling several aspects of metabolism, energy utilization, and cell growth in response to nutrient abundance within the cell. Its hyperactivation usually occurs due to increased metabolic load, being a feature of obesity-related diabetes progression. Additionally, it is also considered a cause of insulin resistance. One of the mechanisms by which mTORC1 inhibits autophagy involves the phosphorylation of the transcription factor EB (TFEB). When mTORC1 is suppressed, TFEB is activated and translocates to the nucleus to up-regulate the expression of genes linked to autophagic and lysosomal production [63]. In a study by Ji et al., high glucose led to enhanced lipid droplet formation, which was accompanied by TFEB activation and reduced autophagy gene expression in rat INS-1 cells. The authors suggest that, under hyperglycemia, lipid clearance is disrupted in β-cells contributing to the glucolipotoxicity-mediated functional impairments in islets [64].

It has been suggested that defects in human IAPP (hIAPP) turnover and cellular processing by proteasome and autophagy may contribute toward toxic hIAPP accumulation and ultimately, β-cell apoptosis. Supporting this, β-cell exposure with hIAPP amplifies autophagosome development and by blocking the autophagic pathway leads to increased hIAPP toxicity [65]. Moreover, in hIAPP transgenic mice, deficient autophagy induced the development of diabetes, which was not observed in animals just expressing hIAPP or lacking autophagy alone. Autophagy impairment promoted the accumulation of hIAPP oligomers and amyloid structures in pancreatic islets of mice, leading to increased death and reduced β-cell mass. Noteworthy, enhancement of autophagy ameliorated the metabolic profile of these animals when fed with a high-fat diet [66]. Accordingly, it was recently shown that MSL-7, a specific autophagy enhancer, diminished hIAPP oligomer accumulation in human pluripotent stem cell-derived β-cells reducing apoptosis. In hIAPP-expressing mice, administration of MSL7 also improved glucose tolerance and β-cell function, which was accompanied by reduced hIAPP oligomer/amyloid accumulation [67]. 

### 1.5. Dysbiosis in Metabolic Dysfunction

As made clear until this point, diabetes is a result of complex and variate factors. In recent years another aspect has been widely discussed as a driving factor in the pathophysiology of diabetes: altered microbiota. The introduction and spread of affordable high-throughput sequencing technologies have allowed researchers to identify the players in the functional analysis of intestinal microbiota [68]. The more abundant information from murine animals suggests that changes in the gut microbiota play a key causal role in the development of diabetes. However, the causality in humans is not yet fully seen and reports seem to be slightly contradicting [69,70]. This may be due to study design, study location, sample preparation, different data analysis, etc. On the other hand, the evaluation of gut microbiota of unhealthy individuals raises questions whether the seen alterations in comparison to healthy populations are prior or a consequence of disease development. The potential molecular mechanisms related to the crosstalk between intestinal microbiota and metabolic diseases such as diabetes have already been reviewed elsewhere [70,71].

The influence between host and gut microbiota is a two-way interaction. The metabolites produced by the microbiota are key mediators of the host glucose metabolism [72,73]. Among the primary end products of bacteria fermentation of polysaccharides are the short-chain fatty acids (SCFA). These contribute to the regulation of lipid and glucose metabolism—they are the primary energy source for enterocytes and colonocytes and impact intestinal inflammation and oxidative stress. Moreover, they stimulate the secretion of glucagon-like peptide (GLP) -1 and -2, adiponectin, and insulin [74,75]. Therefore, SCFAs at low levels in the blood and gastrointestinal tract are associated with diabetes-related dysbiosis [76].

Even though the definition of a healthy microbiota is not clear, there are multiple factors that can cause interindividual variability in the microbiota: age, ethnicity, host genetics, mode of delivery at birth, diet, medication, lifestyle habits, among others. One of the consensuses is that the composition of a healthy individual’s microbiota should be highly diverse. Several metagenomic studies have linked obesity to reduced bacterial gene variety [77,78,79]. Furthermore, evidence shows that environmental and lifestyle factors seem to cause a decrease in diversity [80]. Particularly, the use of antibiotics seems to play a role in changing this balance. For instance, it increases the expansion of the Firmicutes phyla, which has been associated with obesity and diabetes [81,82]. 

One of the most prescribed oral antidiabetic medications, metformin, is known to change gut microbiota composition [83,84]. In fact, evidence from germ-free mice receiving fecal matter transplants from subjects treated with metformin showed better glucose tolerance than those from subjects treated that with placebo [85]. This indicates that the gut microbiota changes are a part of the effectiveness of the drug, which reinforces the importance of microbiota in the disease pathophysiology. 

## 2. (Poly)Phenols in Diabetes

(Poly)phenols are a large group of phytochemicals with largely diverse structures, from simple phenolic acids to complex ellagitannins. They have been the target of vast scientific investigations due to their intrinsic low toxicity and numerous beneficial health effects. In the last years, several studies report the effect of (poly)phenols against a myriad of pathologies, namely cardiovascular diseases, cancer, Parkinson’s, and Alzheimer’s disease, and in particular diabetes [86,87,88,89,90].

### 2.1. Ellagitannins and Ellagic Acid 

Ellagitannins (ETs) are a group of high molecular weight (poly)phenols mainly found in strawberry, raspberry, blackberry, walnuts, and pomegranate [91]. ETs consumption has been consistently associated with positive effects towards many pathologies, including metabolic disorders and diabetes [92,93]. Reports have shown that pomegranate consumption results in improved short-term hypoglycemic responses, pancreatic β-cell function, and insulin resistance in individuals with diabetes [94,95]. The effectors of these responses were thought to be ETs, via punicalagins and punicalins, exerting protection towards blood glucose and insulin resistance when administered in animals [92,93].

Even though multiple health benefits have been attributed to ETs, it is known that they have limited bioavailability [96,97], being unable to reach circulation in considerable concentrations [98]. ETs have been detected in the gastrointestinal tract but not in other organs as they are metabolized mainly in the stomach, but also the small intestine or the colon into ellagic acid (EA) [99,100]. However, EA is also poorly absorbed in the gastrointestinal tract. A study showed that increasing ETs consumption did not improve the amounts of EA in circulation [101]. EA produced from ETs hydrolysis reaches the colon where it is further hydrolyzed and metabolized by the gut microbiota into dibenzopyran-6-one derivatives, known as urolithins [102]. In that view, urolithins should be considered as compounds with bioactivity in response to consumption of ET- or EA-rich foodstuffs (Figure 1 and Figure 2A). 

### 2.2. Urolithins and Metabotypes

Even though it is known that EA metabolism by the gut microbiota originates the urolithins, it has been observed a significant interindividual variability regarding the type of urolithin produced in the human body. In other words, not all individuals produce the same types and the same relative amounts of urolithins in response to ET or EA exposure. Since urolithin production is mainly dependent on the gut microbiota, the highly variable composition of the microbiota of each individual impacts directly the presence and/or availability of different urolithins [103].

The capacity of the human body to metabolize EA into different urolithins can be stratified into metabotypes and the classifications have changed over time. The most recent one classifies metabotypes in A, B, and 0: metabotype A (MetA) produces urolithin A and their conjugates; metabotype B (MetB) produces urolithin B, isourolithin A, and urolithin A; and metabotype 0 (Met0) does not produce any urolithins in detectable amounts (Figure 1). Furthermore, the interplay between (poly)phenols and microbiota works in two ways, meaning that microbiota affects metabolites production as well as (poly)phenols consumption also modulates the microbial population of individuals. Thus, a subpopulation of Met0 can change to MetA or MetB after long-term and/or high exposure to ETs or EA [104].

In terms of the prevalence of metabotypes, the numbers seem to be affected by different aspects. One study evaluated a large cohort of Caucasian individuals and showed that the prevalence of metabotype is affected by age. With aging, the percentage of MetA decreases, and MetB increases, leading to reduced MetA/MetB ratios until 30–40 years of age, which then stabilizes. On the other hand, ~10% percentage of Met0 seems to be unaltered across the analyzed ages (5–90 years of age) [105]. The stratification of participants by metabotype in intervention studies may be the key to decreasing the high variability of results among studies reporting (poly)phenol consumption effects. Furthermore, there has been an interest in relating metabotype, individuals’ characteristics, and disease risks. A study suggested that in individuals with diseases related to gut dysbiosis (metabolic syndrome or colorectal cancer), the prevalence of MetB increases [104], but this was only proven for adults. The association was lost when larger age intervals were taken into consideration [105]. In addition, no relation between metabotype and gender, medication, or adherence to the Mediterranean diet was seen [105]. One report comparing normal weight, overweight-obesity, and metabolic syndrome individuals showed that obese subjects capable of producing urolithin A, isourolithin A, and/or urolithin B were at higher cardiovascular risk (calculated based on lipid profiles and plasma glucose) than obese individuals who produced only UroA [102].

#### 2.2.1. Phase-II Metabolism

Urolithins are further modified by large intestine enterocytes and the liver, in phase-II metabolism, suffering the addition of chemical moieties, usually sulphate or glucuronide groups. In humans, these are the urolithin forms detected in the highest concentrations, indicating active phase II metabolism and enterohepatic recirculation, as depicted in Figure 2A [106].

Even though phase-II urolithin metabolites are often found at higher concentrations than the respective aglycones [106,107], these compounds have lower bioactivity than their deconjugated counterparts, for instance in cancer anti-proliferative activity and inflammation [108,109]. On the other hand, urolithin metabolites resulting from glucuronidation, commonly referred to as the one with the highest circulating concentrations, are susceptible to glucuronidase. This is extremely relevant since glucuronidase is highly expressed in high inflammation areas, meaning that the aglycone compounds may also play a role on their own [110,111]. It was shown that inflammation caused by lipopolysaccharide in rats leads to tissue deconjugation of urolithin A-glucuronide to urolithin A, showing that in vitro studies with aglycone are relevant and valid [111].

#### 2.2.2. Bioavailability 

Urolithins have been encountered in different organs in detectable concentrations, both in animal models and in humans. The most frequently found are urolithin A, isourolithin A, and urolithin B, and their phase II conjugates. These compounds are bioavailable, reaching the plasma and being excreted in urine [112]. 

A study provided a single pomegranate juice concentrate containing 387 mg/L of anthocyanins, 1561 mg/L of punicalagins, 121 mg/L of ellagic acid, and 417 mg/L of other hydrolyzable tannins to 18 human subjects. On the day following juice consumption, urolithin A-glucuronide was found in urine samples of 16 subjects and urolithin B-glucuronide in samples of five subjects. As for plasma, in the seven subjects tested for EA metabolites, the maximum concentrations of total urolithins 6 h after consumption were 0.14 μmol/L and 0.01 μmol/L for urolithin A and urolithin B, respectively [112]. Another study provided a larger dose of pomegranate juice (1 L per day, for 5 days), containing 5.58 g/L of (poly)phenols, including 4.37 g/L of punicalagin isomers. In this report, the total circulating urolithins were estimated to reach up to 18.6 mmol/L, with high inter-individual variability [98].

In terms of human and animal studies using pure compounds, urolithin A is by far the most common. In a safety study, urolithin A was administered as a soft gel to elderly adults, being then found in circulation in all subjects, receiving doses from 250 to 1000 mg for 28 days. All urolithin A detected forms (aglycone, glucuronide, and sulphate) had identic kinetics: maximum concentration reached at 6 h (Tmax) with half-life (t_1/2_) values of 17 to 22 h for urolithin A and urolithin A-glucuronide, and 25–58 h for urolithin A-sulphate, and all were eliminated between 72–96 h after the last dosing. There was a dose-dependent increase in total urolithin A steady-levels between 250 and 1000 mg but no difference in the accumulation of urolithin A between single or multiple dosing [106]. 

As for target tissues, methyl urolithin A, urolithin A-sulfate and urolithin A-glucuronide were found in the prostate, intestine, colon, liver, kidney, and lung of rats after oral administration of urolithin A [107,113]. Another study using rats receiving intravenously a bolus of 2.7 μmol containing 23 (poly)phenol microbial metabolites showed that urolithin B was sequestered (and/or metabolized) more rapidly than urolithin A in the tissues and it accumulated preferentially in the heart [114]. Furthermore, a study revealed the presence of aglycones and sulphate forms of urolithin A and urolithin B in the pancreas, liver, and heart of diabetic rats [115]. This data is encouraging regarding the bioavailability of urolithins and their use towards their use in diabetes management. 

Data are scarce on the concentration of urolithins in the organs of humans. One report evaluates 63 individuals with either benign prostate hyperplasia or prostate cancer, consuming 35 g of walnuts or 200 mL of pomegranate juice daily for 3 days. Regardless of the (poly)phenol source, urolithin A-glucuronide was the main found metabolite. However, a high number of prostate samples had no traces of metabolites probably due to the individuals’ fasting before surgery [107]. A similar study provided two capsules of 600 mg polyphenol from pomegranate extract for 2 to 4 weeks, or placebo, before radical prostatectomy, where prostate samples were collected. Even though urolithin A was found in 22 of the 33 prostate samples from the intervention group, there was no difference in the primary outcomes between arms [116]. On the other hand, 52 individuals with colorectal cancer received daily 900 mg pomegranate extract capsules with either high or low punicalagin:EA ratio for 15 days before surgery. Twelve urolithin derivatives were found in the colons of these individuals, with high variability between them. Interestingly, the capsules with high punicalagin content hampered urolithins formation [117]. Finally, a urolithin A safety study following the intake of a 2 g single dose detected the metabolite in the skeletal muscle of individuals after 8 h. Among them, two out of six had traces amounts of urolithin A-glucuronide but no urolithin A-sulphate was detected in any sample [106]. 

Even though the detection of urolithins in circulation and target tissues following the intake of ETs/EA rich sources represents encouraging data for the exploitation of urolithins as therapeutic tools, several aspects need to be considered. Studies reporting the presence and concentration of urolithins and their metabolites, in circulation and tissues, are highly dependent on the type of administration, dose, frequency as well as the time they remain in circulation, and their accumulation in the tissues, among other factors. Furthermore, the type of response to the intake is indeed influenced by inherent factors of the subjects, namely their sex, age, microbiota, health status, etc. For these reasons, although the health claims attributed to urolithins are extremely promising, additional studies are required to support their use in therapeutic approaches.

## 3. Biological Activity against the Pathological Processes of Diabetes

Several studies report the health benefits of urolithin A and urolithin B, aglycone, and phase-II metabolites, towards a multitude of diabetes complications as summarized in Figure 2B.

Chronic hyperglycemia, a common feature of all forms of diabetes, causes a myriad of effects in the human body. Urolithin A treatment on mice fed with a high-fat diet (HFD) has shown improvements in blood glucose and insulin, glucose tolerance, and insulin sensitivity [118]. In contrast, another report using an HFD and high sugar diet showed different results, with decreased fasting blood glucose by urolithin A but no effect on body weight, visceral fat, or liver lipid content [119]. These disparities in the results may result from the differences in the animal strain, the type of diet fed to the animals, the different doses, type of administration, and duration of treatment with urolithin A. Moreover, urolithin A and urolithin B have been shown to prevent AGE, a common deleterious consequence of chronic high blood glucose [120,121]. 

Besides alterations in carbohydrates metabolism, individuals with diabetes also present defects in lipid and protein metabolism. A study looking at serum lipidemic profiles of individuals with metabolic syndrome, which received 8-week berry mixture supplementation containing ETs, showed a moderate response to the ET-rich diet in phosphatidyl cholines, triacylglycerols, phosphatidyl ethanolamines, and cholesteryl esters in the intervention group, but no effect in plasma fatty acid composition when compared to the control group [122]. Furthermore, administration of 40 g of concentrated pomegranate juice to individuals with diabetes for 8 weeks showed a significant reduction in total cholesterol, LDL-cholesterol, LDL-cholesterol/HDL-cholesterol ratio, and total cholesterol/HDL-cholesterol [123]. The effects of urolithin B were also tested in an apoE^−/−^ mice model, showing decreased lipid deposition in the aorta. This study also showed that both urolithin B and its sulphated form caused the increase in cholesterol efflux in macrophages cell lines, a predictor of all-cause and cardiovascular mortality in individuals with coronary artery disease [124]. Since lipid metabolism abnormalities are common in individuals with diabetes, improvements in these different markers caused by urolithin-rich interventions are a good argument for their effects in diabetes management.

Sedentarism and obesity are key factors associated with diabetes, thus compounds contributing to obesity control may represent important adjuvants in diabetes management. The potential of urolithins to fight obesity is evidenced by several studies. Urolithin A, urolithin B, and urolithin C were shown to significantly reduce the accumulation of triglycerides and to increase the oxidation of fatty acids in human adipocytes and hepatocytes [125,126], to induce browning of white adipocytes and to activate lipolysis [127]. Urolithin A also reduced triglyceride build-up in murine adipocytes [128], and bodyweight and fat mass in HFD-fed mice (with no changes in food intake or fat absorption) and in genetically obese mice with reduced food intake [118]. Moreover, both urolithin A and urolithin B decreased body weight and visceral adipose tissue caused by HFD in rats. These effects were accompanied by decreased serum cholesterol, triglycerides, and HDL, and reduced hepatic lipids and liver oxidative stress [129]. Taken together, the evidence seems to indicate that urolithins cause changes in lipid metabolism, leading to changes in body composition and fat deposition. Since obesity is a key driver for diabetes development, these effects are of utmost relevance for their case as diabetes therapy potential hits.

Inflammation is a central pathway in the pathological processes causing diabetes [26]. Existing evidence has shown the effects of urolithins towards the prevention of inflammatory responses to high glucose in rat cardiomyocytes by urolithin B and urolithin B-glucuronide [130], and the recovery of hemodynamic performance parameters, cellular mechanical properties, and calcium dynamics in streptozotocin (STZ)-induced diabetes rat model by urolithin B and urolithin A in a lesser extent [115]. In a study evaluating the effect of urolithins on lipopolysaccharide (LPS)-stimulated macrophages, urolithin A was the most effective inhibiting inflammatory responses, stimulating the expression of anti-inflammatory cytokines, and decreasing NO production and NO synthase protein expression [131]. 

The mechanisms by which IAPP dysregulation damages β-cells are beginning to be understood [65,132,133], and the search for compounds overcoming its toxic effects continues to be a hot topic. Evidence of (poly)phenols hindering IAPP aggregation and protecting β-cells are emerging, as reviewed in [134,135]. Particularly, 1,2,3,4,6-pentagalloyl glucose (PGG), a compound present in pomegranate and the precursor of ETs, has been reported as a potent inhibitor of IAPP aggregation [136]. Concerning urolithins, it was shown that urolithin A significantly decreased amyloid-beta deposition, ameliorates cognitive impairment and neuronal apoptosis [137,138]. In SH-SY5Y neuroblastoma cells and iPS-derived neuronal differentiated cells, urolithin A reversed the levels of Abeta1-42, APP, and BACE1 caused by the high glucose [139]. 

Another reported effect of urolithins involves the induction of autophagy. This effect is extremely relevant in conformational diseases, including diabetes [59]. One study showed striking urolithin A-mediated protection in a mice model subjected to STZ injection regarding blood glucose, inflammation, oxidative stress, islet architecture, and apoptosis, with the near abolition of protection when the autophagy inhibitor chloroquine was added to the treatment [140,141]. This shows the power of urolithin A in preventing common pathological effects of diabetes and how autophagy is preponderant as a mechanism of action. Urolithin A also extends the lifespan and fitness of *C. elegans* through stimulation of mitophagy, the selective autophagy for the elimination of damaged mitochondria. The same effect was seen in the muscle of both young and old mice, resulting in better exercise performance [142]. The mitochondria were also implicated in the positive effects of urolithin A in SH-SY5Y cells, in which hyperglycemia causes mitochondrial calcium accumulation and increased mitochondrial reactive oxygen species (ROS), effects that urolithin A ameliorated [139]. 

Lastly, safety study with urolithin A on sedentary elderly individuals [106]. Remarkably, urolithin A caused significant changes in molecular markers of mitochondrial health, particularly short-chain acylcarnitine, which indicates improved fatty acid oxidation in skeletal muscle after 8 h after a single take of 500 or 1000 mg dose. Furthermore, these doses led to increased transcriptional levels of mitochondrial genes, and stimulation of mitochondrial biogenesis. These data suggest the activation of mitophagy in the skeletal muscle of the volunteers, as reported earlier in animal models.

Urolithin Use as a Therapeutic Approach

Since data on humans are only available in sedentary elderly individuals, there is a clear need for further investigations in different populations. In this view, it would be of great importance to stratify the studied population, besides the common criteria of age, sex, health status, etc., regarding their metabotype. This is relevant since it may significantly alter the results based on the individual’s capacity to produce a certain urolithin. The stratification of the population based on their metabotype has been a target of recent investigations, with several challenges both conceptual and technical (as reviewed in [143]). 

Another possible confounding factor of population response to a certain diet intervention is the medication. As it is known, metformin is the most used oral antidiabetic medication, and it has a described effect in altering the gut microbiota. In fact, it has been seen that metformin use is a strong confounding factor in a metagenomic analysis [84]. For this, metformin use also must be taken into consideration when stratifying populations and evaluating the effects of dietary interventions. 

Nevertheless, the metabotype stratification is of higher relevance if urolithins are to be ingested through food matrices, i.e., from a nutritional point of view. In pharmacological approaches, aspects other than the metabotype must be taken into consideration regarding the response to the compound, namely bioavailability. There is only one study that has evaluated the safety of urolithin A intake in human subjects, showing that it is safe until 1000 mg for 28 days [106]. Since this is the only available data, further research is necessary to evaluate the safety of urolithin A and B, at different doses and prolonged times. Even though the authors found no signal of compound accumulation with this experimental design, longer setups are required to draw defined conclusions.

## 4. Conclusions

Several reports on the role of urolithins against diabetes suggest their activity as anti-inflammatory, anti-obesity, and cytoprotective agents, among others. Such compounds with protective effects towards different processes associated with diabetes may be a powerful weapon to tackle such a complex disease.

Several studies report the bioactivity of urolithins, or their phase-II metabolites, stating the need to focus on the effects of each compound alone. However, it is also important to investigate the putative effects of combining different urolithins as some are expected to be present simultaneously in the body depending on the metabotypes. 

Concerning the physiological concentrations of these compounds, it has been widely discussed that most in vitro studies use urolithin concentrations beyond those found in circulation after consumption of ET-rich foodstuffs. However, it must be taken into consideration that the metabolite’s tissue distribution and metabolism have not yet been fully understood, particularly in tissues with inflammatory responses.

For all these reasons, at this point, human intervention studies are necessary and urgent to validate the effects found in vitro. Furthermore, it is probable that delivery strategies need to be coupled to the interventions to overcome the inherent difficulties caused by the different metabotypes or reduced bioavailability to exert an effect on target tissues. Regardless of the path that must be taken to understand deeper the intricacies of urolithins bioactivity towards diabetes, it is indisputable that the existing knowledge designates these small molecules as strong candidates for the fight against diabetes.

## Figures and Tables

**Figure 1 nutrients-13-04285-f001:**
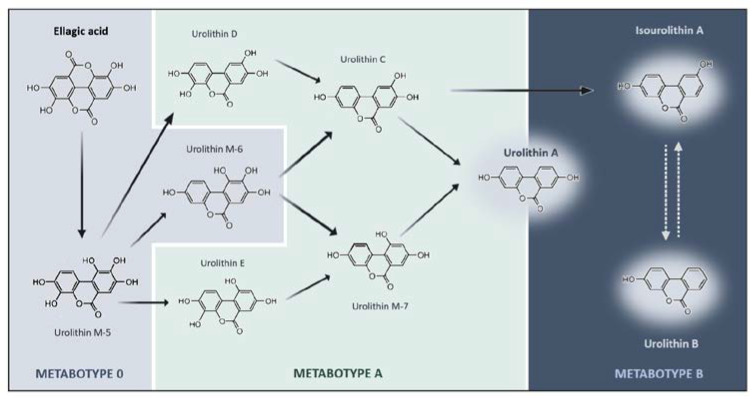
**Schematic representation of ellagic acid catabolic pathway by colonic microbiota and the respective urolithin metabotypes**. Ellagic Acid (EA) is metabolized into bioactive metabolites through lactone-ring cleavage, decarboxylation, and dihydroxylation reactions. Depending on the individual’s microbiota composition, EA can be metabolized into distinct types of urolithins. Such differences can be stratified into three metabotypes (A, B, and 0), depicted by specific color codes. Metabotype 0 (gray) does not produce the final urolithins at least in detectable concentrations. Metabotype A (yellow) is characterized by the production of urolithin A and its conjugates. Metabotype B (green) is characterized by the production of urolithin B, urolithin A, and isourolithin A. The chemical structures represented indicate the intermediates and final metabolites that are produced along the different pathways of ellagic acid metabolism.

**Figure 2 nutrients-13-04285-f002:**
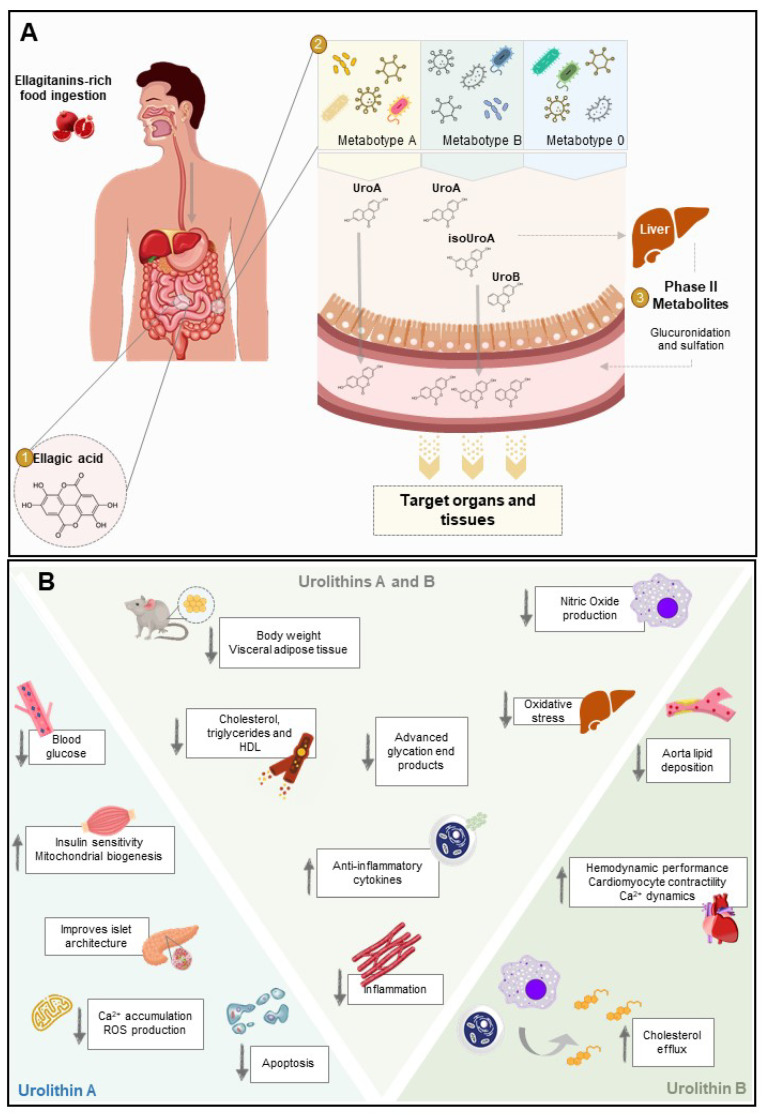
**Metabolism of ellagitannins to yield bioactive urolithins with potential action for diabetes.** (**A**) Several dietary products (e.g., pomegranate) are rich in ellagitannins (ETs), a class of natural (poly)phenols. Upon ingestion, ETs are hydrolyzed in the small intestine into ellagic acid (EA), reaching the colon. Through the action of microbiota, EA is converted to urolithins depending on the individual’s metabotype. In phase-II metabolism, urolithins are further modified by large intestine enterocytes and the liver (methylation, glucuronidation, and sulfation) to yield conjugated forms of these metabolites. Both conjugated and deconjugated urolithins are able to enter circulation and reach the target tissues where they will perform their function. Although conjugated forms are often found at higher concentrations than the respective deconjugated counterparts, aglycones show much higher bioactivity. (**B**) Once absorbed by the target tissues, urolithin A and urolithin B may exert cell-specific activities that confer protection towards a multitude of diabetes complications. In mice, urolithin A decreases blood glucose and increases glucose tolerance and insulin sensitivity. It also protects against islet architecture disruption, oxidative stress, and cell death. Urolithin B is described to reduce lipid deposition in the aorta and increase cholesterol efflux on macrophages cell lines. In adult Wistar rats with streptozotocin-induced diabetes, urolithin B prevents the negative impact of altered diabetic milieu on cardiac performance. Regarding the common effects, urolithin A and urolithin B influence lipids levels. They reduce body weight and fat mass in high fat diet-fed mice and decrease the amounts of cholesterol, triglycerides, and high-density lipopolysaccharide (HDL) in serum. Urolithin A and urolithin B also protect cells from inflammatory response and oxidative stress.

## Data Availability

Not applicable.

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
