# Peer review of "Urolithins: Diet-Derived Bioavailable Metabolites to Tackle Diabetes"

_nutrients, 2021, doi:10.3390/nu13124285_

Round 1

Reviewer 1 Report

The submitted review summaries the hitherto studies on gut microbiota derived postbiotic metabolites in the context of their potential anti-diabetic activity. In my opinion the review is very well written and expresses critical approach on included data as well as gives an interesting perspective on further studies. 

Author Response

We would like to acknowledge the time and effort dedicated by Reviewer 1 in reading our manuscript. We would also like to recognize the appreciative words regarding our review.

Reviewer 2 Report

The manuscript "Urolithins: diet-derived bioavailable metabolites to tackle diabetes" offers a novel and exhaustive overview on the possibility to use polyphenols integration in the diet of diabetic patients. Apart from some details, the review reads well and proposes an interesting actual topic, which is also shared in the cancer field (10.3389/fnut.2021.647582, possibly to be added in line 252). A further improvement in transmitting the novelty of the work could be achieved by making paragraph 2.3 as an independent chapter, where the potential applications of urolithins in clinical settings can be discussed in a more detailed way, including advantages, limits, and current challenges and critical point to overcome. 

Overall, the manuscript is already very well presented. Further minor comments are listed below:

  1. Regarding the diabetes chapter, many complex signaling pathways and factors are described and detailed. Although Figure 2 gives a clear overview on how urolithins impact on diabetes, the connections with these pathways and factors (e.g., Urolithins and PI3K) previously described are not fully mentioned. This information could be introduced as a table, a figure, or in the text, if details on mechanism of action on the discussed pathways (and factors) in diabetes are available. It must be additionally checked if, even if not studied in the context of diabetes, there are available results on common pathways altered in other diseases (e.g., other metabolic disorders, cancer, etc.).
  2. As it is discussed in the chapter about polyphenols, it may be of interest to implement (e.g., in the chapter about diabetes) also the recent findings about the connection between gut microbiota and diabetes. Are there other metabolites together with urolithins with potential action on diabetes depending on gut microbiota? This information could enrich further the interesting aim of stratifying patients in relation to the urolithins metabotypes, as a reflection also of the individual gut microbiota activity. 
  3. Line 249: investigations instead of investigation
  4. Line 329: leads instead of lead
  5. Line 388: The authors comment on the limiting aspects to be considered about the variability of urolithins bioavailability. Should this topic be discussed as part of paragraph 2.3? The authors could add also comments about which steps must be achieved to support their use in therapeutic approaches, e.g., characterization of physiological parameters, including microbiota, and correlation with the various urolithins metabotypes.
  6. Line 439: Figure 2A should be introduced already in section 2.1.1.
  7. Paragraph 2.3. is the main paragraph of the review, and should be intended more as an independent chapter. Here, the use of urolithins in clinical settings could be further discussed in a critical way (connection to comments 1 and 5). Over advantages, limits must be discussed, possibly also in relation to the technical detection of these metabotypes, if necessary.
  8. In Figure 2, it may be necessary to make clearer the meaning of "improves blood glucose", maybe in "decreases blood glucose".
  9. Regarding the modality of action in diabetes of urolithins in Section 2.3, it may be interesting to add more informations on how using different urolithins metabotypes may target different sub-groups of patients, if data are available. In connection with previous comments (1, 5, and 7), it would be clearer to have a separate chapter treating the limits, advantages, and current challenges of therapeutic applications of urolithins.
  10. It is necessary to revise the hyphenations along the complete text.

Author Response

The manuscript "Urolithins: diet-derived bioavailable metabolites to tackle diabetes" offers a novel and exhaustive overview on the possibility to use polyphenols integration in the diet of diabetic patients. Apart from some details, the review reads well and proposes an interesting actual topic, which is also shared in the cancer field (10.3389/fnut.2021.647582, possibly to be added in line 252). A further improvement in transmitting the novelty of the work could be achieved by making paragraph 2.3 as an independent chapter, where the potential applications of urolithins in clinical settings can be discussed in a more detailed way, including advantages, limits, and current challenges and critical point to overcome. 

We would like to acknowledge all the insight provided by Reviewer 2 regarding our work and the provided suggestions to improve the manuscript. As suggested, the paragraph 2.3 was transformed into an independent chapter, with improvements in terms of discussing the application of urolithins in a therapeutic approach. Furthermore, the suggested reference was added in the proposed line.

Overall, the manuscript is already very well presented. Further minor comments are listed below:

  1. Regarding the diabetes chapter, many complex signaling pathways and factors are described and detailed. Although Figure 2 gives a clear overview on how urolithins impact on diabetes, the connections with these pathways and factors (e.g., Urolithins and PI3K) previously described are not fully mentioned. This information could be introduced as a table, a figure, or in the text, if details on mechanism of action on the discussed pathways (and factors) in diabetes are available. It must be additionally checked if, even if not studied in the context of diabetes, there are available results on common pathways altered in other diseases (e.g., other metabolic disorders, cancer, etc.).

We would again like to appreciate the insight provided in the comments by the Reviewer. In fact, Figure 2 describes in a summarized way the pathways involved in urolithin’s mode of action towards diabetes. Along the text, these pathways are explained in further detail, when information is available in the literature, and we feel that including that information in another support, such as a table or another figure would only add to repetition. Furthermore, regarding the inclusion of urolithin’s action on pathways from other diseases that are common to diabetes, we had already tried to include that whenever we felt information in diabetes was scarce (for example, with Alzheimer’s data in lines 547-551). Naturally, with diabetes being such a complex interplay of factors, we are sure that more details could be added by inserting parallel works in other diseases but that could deviate from our goal and scope.

  1. As it is discussed in the chapter about polyphenols, it may be of interest to implement (e.g., in the chapter about diabetes) also the recent findings about the connection between gut microbiota and diabetes. Are there other metabolites together with urolithins with potential action on diabetes depending on gut microbiota? This information could enrich further the interesting aim of stratifying patients in relation to the urolithins metabotypes, as a reflection also of the individual gut microbiota activity. 

As suggested by Reviewer 2, a section on the relation between the microbiota and diabetes has been included in Section 1. Since we want to keep the focus of our review on urolithins, we consider it more prudent not to divert the reader’s attention to other (poly)phenol metabolites. Nevertheless, some considerations regarding microbial metabolites which influence the host’s metabolism were included.

  1. Line 249: investigations instead of investigation - The correction was made to the text.

  1. Line 329: leads instead of lead - The correction was made to the text.

  1. Line 388: The authors comment on the limiting aspects to be considered about the variability of urolithins bioavailability. Should this topic be discussed as part of paragraph 2.3? The authors could add also comments about which steps must be achieved to support their use in therapeutic approaches, e.g., characterization of physiological parameters, including microbiota, and correlation with the various urolithins metabotypes.

A section (3.1) was added regarding the use of urolithins as therapeutics, both from nutritional and pharmacological point of view. In this section, the limitations of the available data, as well the necessary steps to take in the future were elaborated on.

  1. Line 439: Figure 2A should be introduced already in section 2.1.1.

 The figure was introduced in the suggested section. Also, throughout the test, it was clarified whether part A or part B of Figure 2 should be focus on.

  1. Paragraph 2.3. is the main paragraph of the review, and should be intended more as an independent chapter. Here, the use of urolithins in clinical settings could be further discussed in a critical way (connection to comments 1 and 5). Over advantages, limits must be discussed, possibly also in relation to the technical detection of these metabotypes, if necessary. - The section 2.3 was transformed in section 3 as to give it the deserved focus.

  1. In Figure 2, it may be necessary to make clearer the meaning of "improves blood glucose", maybe in "decreases blood glucose". – The correction was made to the text.

  1. Regarding the modality of action in diabetes of urolithins in Section 2.3, it may be interesting to add more informations on how using different urolithins metabotypes may target different sub-groups of patients, if data are available. In connection with previous comments (1, 5, and 7), it would be clearer to have a separate chapter treating the limits, advantages, and current challenges of therapeutic applications of urolithins.

Regarding comments 5, 7 and 9, a sub-chapter was added discussing exclusively the use of urolithins in a therapeutic approach, the limitations of current data and the necessity to further explore the subject. In addition, a recently published review was cited regarding the limitations of population stratification and the state of the art regarding the technical detection of the metabotypes (https://pubs.rsc.org/en/content/articlelanding/2021/FO/D1FO02033A).

  1. It is necessary to revise the hyphenations along the complete text. – The text was reviewed accordingly.